# Rationale, Design and Methods Protocol for Participatory Design of an Online Tool to Support Industry Service Provision Regarding Digital Technology Use ‘with, by and for’ Young Children

**DOI:** 10.3390/ijerph17238819

**Published:** 2020-11-27

**Authors:** Susan Edwards, Andrea Nolan, Michael Henderson, Susan Grieshaber, Kate Highfield, Andi Salamon, Helen Skouteris, Leon Straker

**Affiliations:** 1Institute for Learning Sciences and Teacher Education, Australian Catholic University, Melbourne 3065, Australia; Suzy.Edwards@acu.edu.au; 2School of Education, Faculty of Arts and Education, Deakin University, Geelong 3217, Australia; a.nolan@deakin.edu.au; 3Faculty of Education, Monash University, Melbourne 3800, Australia; michael.henderson@monash.edu; 4School of Education, La Trobe University, Melbourne 3086, Australia; S.Grieshaber@latrobe.edu.au; 5Early Childhood Australia, Canberra 2600, Australia; KHighfield@earlychildhood.org.au; 6School of Teacher Education, Charles Sturt University, Bathurst 2795, Australia; asalamon@csu.edu.au; 7Monash Centre for Health Research and Implementation, School of Public Health and Preventive Medicine, Monash University, Melbourne 3800, Australia; helen.skouteris@monash.edu; 8Warwick Business School, The University of Warwick, Coventry CV4 7AL, UK; 9School of Physiotherapy and Exercise Science, Curtin University, Perth 6845, Australia

**Keywords:** young children, technology, digital practices, relationships, health and wellbeing, digital citizenship, digital play, service provision

## Abstract

Adults who educate and care for young children are exposed to mixed-messages about what is in the best interests of young children in digital society. Such mixed-messaging makes adult decision-making about technology use in the best interests of young children hard to achieve. This project addresses this problem by working with leading organisations providing services related to quality digital media production, online-safety education, digital play and digital parenting. Using a Participatory Design approach, families, educators, industry partners and researchers will conduct mixed-methods investigations concerning: Relationships; Health and Well-being; Citizenship; and Play and Pedagogy to identify practices concerning technology use ‘with, by and for’ young children. Iterative design cycles will develop an Online Tool to support organisations providing services to young children and the adults responsible for their education and care. As society becomes more digital families and educators need new knowledge about what people do in digital society to inform their decision-making. This project will support organisations to use an empirically informed approach to service provision regarding using technologies in the best interests of young children.

## 1. Introduction

The industry partners on this project face a common problem. How can they most effectively advise and engage young children, their families and educators in digital technology use that is in the best interests of young children? For the purpose of this project, young children are defined as those aged birth to six years. The industry partners on this project are involved with young children, their educators and families from a multitude of perspectives, including the production of quality digital content for young audiences, online-safety education and protection, supporting parents with digital parenting and enabling educators to provide young children with opportunities for digital play in early childhood education and care settings. There are seven industry partners on this project, including: the Australian Broadcasting Corporation, the Australian Federal Police, the Alannah & Madeline Foundation, Deeper Richer, Early Childhood Australia, the eSafety Commissioner, and the Raising Children Network.

These industry partners join with the researchers to investigate the ranges of practices enacted and shared amongst children, families and educators within digital society. It is now well accepted that young children are growing up in digital society, using networked technologies for multiple purposes [1]. This includes, communicating with friends and family, entertainment, learning, recreation and play. During COVID-19, many young children and their families were particularly reliant on using technologies to remain in contact with extended families, for remote learning, and entertainment. As using technologies has become a more common place aspect of daily life for young children and their adults, attention has been directed towards advising families and educators on the most appropriate technology use in the early years. This has resulted in a series of recommendations, that are often conflicting in their suggestions, such as limiting screen time [2,3] versus regulating technology use according to family circumstances [4]. Conflicting guidelines make it difficult for organisations serving young children, their families and educators, such as the Industry Partners on this project to make informed decisions about the content they develop, the advice they provide, and the supporting materials, resources and programmes they generate for their target audiences regarding technology use in the best interests of young children. Already proving problematic prior to COVID-19, conflicting recommendations are further highlighted as an issue for the Industry Partners during COVID-19 as a period during which young children and their adults relied extensively on digital technologies for health care (e.g., telehealth), education and socialisation. Thus, this project sets up to identify the practices in the best interests of young children growing up in digital society. The identified practices will be converted into digital exemplars embedded in an online tool, accessible to the industry partners as an interoperable database upon which they may draw to inform their own service provision working with children, families and educators.

In this project, digital society is understood as a function of human innovation in the invention and use of digital technologies by people for generating, storing, sharing and communicating information between people and objects [5]. Research shows that young children and their adults are active participants in digital society, with infants, toddlers and pre-schoolers using touchscreen devices, Internet of Toys, wearables and voice-activation [6]. Young children regularly consume digital content, play digital games, engage with apps and video-conference family and friends. Young children are also implicated in the digital networks of the adults who educate and care for them. This occurs through ‘sharenting’ [7], and via the digital documentation of their learning by educators [8]. Because digital society is a function of human innovation in the design and use of technologies by people it may be considered in terms of practices. Practices give rise to social situations through the combination of actions and interactions in which people participate daily [9]. To date, little attention has been paid to the range of practices enacted and shared amongst children and their adults as participants in digital society. This project therefore focuses on the identification of practices in the best interests of young children growing up in digital society.

Practices in digital society encompass those involving both young children’s health and wellbeing, and their education and developmental outcomes. This project therefore involves the Industry Partners and researchers in an interdisciplinary investigation directed towards achieving new knowledge about those practices in digital society that are in the best interests of the whole child working from a combined health and education perspective.

### 1.1. Conceptual Framework

This project takes a novel approach to identifying practices enacted and shared amongst children and adults in digital society by integrating practice theory and critical constructivism to provide the conceptual framework.

Practice theory explains practices as the actions and interactions shared amongst people over time, with this process of sharing creating the society in which they participate [10]. Practices are defined as actions and interactions [9], and actions and interactions are identifiable as ‘doings’ and ‘sayings’ [11]. Practices are enacted and shared by people across diverse circumstances. This occurs according to the value different doings and sayings hold for people within their situations, such as children, families and educators playing and working in diverse settings, such as Family Day Care (provided by caregivers in home settings), Playgroup (attended by children and families for play and socialisation opportunities up to two hours per week), Long Day Care (provided by early childhood education and care services for children aged 8 weeks to five years) and/or Kindergarten (provided by 4-year degree qualified educators for children aged 4–5years for fifteen hours per week). When practices are shared across situations a field of action intelligibility is created [10]. The field is an in-situ representation of the actions and interactions comprising society.

Critical constructivism argues against the technological determinist idea of technologies as socially causative and consequently impacting on children’s learning and development [12]. Instead, critical constructivism suggests that all technologies are invented and used according to human values, with values therefore shaping how practices are enacted and shared amongst people in digital society [13]. This means digital society can never be value-neutral. People create their own “paths of progress” [14] characterizing valued technology use according to the situations in which they participate. Praxis, in the best interests of young children in digital society, is therefore empirically identifiable according to the field of action intelligibility, which defines valued technology use amongst young children, their families and educators in diverse situations.

This project uses the Early Childhood Australia (ECA) Statement on Young Children and Digital Technologies [15] to stimulate the enactment and sharing of practices amongst children, families and educators as those indicated of value to them in diverse situations, including Family Day Care, Playgroups, Long Day Care and Kindergartens. Five of the industry partners on this project contributed to a national digital policy group responsible for producing the ECA Statement. These organisations were the Alannah and Madeline Foundation, Early Childhood Australia, Deeper Richer, the eSafety Commissioner and the Raising Children Network.

The ECA Statement provides practice-advice for educators and families concerning technology use ‘with, by and for’ young children growing up in digital society in terms of four main areas of known importance for young children’s health and education. These areas are: (1) Relationships; (2) Health and Wellbeing; (3) Citizenship; (4) Play and Pedagogy. The practice-advice provided in the ECA Statement is informed by a strong reading of the literature concerning the optimal use of technologies ‘with, by and for’ young children, including aspects such as relationships with significant others, sleep, physical activity and posture, online-safety and digital rights, and young children’s digital play [16]. The ECA Statement clearly acknowledges that young children’s participation in, and experience of, digital society is subject to variation according to socio-economic status, cultural and linguistic diversity, gender and geographic location [15].

### 1.2. Project Aim

This project aims to support decision making by adults who educate and care for young children growing up in a digital society. To promote technology use that is in the best interests of young children the project will involve the industry partners collaborating directly with children, families and educators, working alongside the researchers. A practical outcome of the project will be an online tool for use by the industry partners, incorporating a shared database of practices to support empirically informed service provision for their target audience (adults who care for and educate young children).

## 2. Methods

### 2.1. Objectives

The project will use a collaborative Participatory Design process [17] to accomplish two objectives:(1)identify the practices enacted and shared amongst children, families and educators in digital society,(2)create an online tool for empirically informed service provision about digital technology use ‘with, by and for’ young children.

### 2.2. Study Design

The participatory design approach in this project will involve researchers, industry partners and end-users (e.g., children, families and educators) collaborating on a problem with the intention of creating a solution [18] (in this case, practices which can support young children growing up in a digital society).

Participatory design operates on repeating cycles of design and redesign. Each cycle comprises three phases. These are: (1) Exploring; (2) Discovering; and (3) Prototyping [17].

In this project one cycle of design [for enacting practices] and a further cycle of redesign [for sharing practices] will be conducted. Both the design and redesign cycles are conducted across four related Investigations-one per area of the ECA Statement. The Investigations therefore canvass: (1) Relationships; (2) Health and Wellbeing; (3) Citizenship; and (4) Play and Pedagogy.

Each investigation uses a defined method, with the combination of methods across all four Investigations reflecting the use of integrated methods in participatory design more generally [18]. Combining qualitative and quantitative methods is also considered ‘optimal’ for research concerning digital practices in childhood [19]. The methods used for each Investigation are: Ethnography [20] for Relationships; Longitudinal [21] for Health and Wellbeing; Quasi-experimental [22] for Citizenship; and Intrinsic Case Study [23] for Play and Pedagogy.

Each Investigation addresses a sub-research question of the primary research question: “What practices do children, families and educators enact and share according to the four areas of practice-advice provided in the ECA Statement on Young Children and Digital Technologies?”. The sub-questions are designed to capture the remit of practice-advice provided for each area in the ECA Statement, whilst also reflecting current concerns in the literature regarding young children and digital technologies. For Relationships, this includes peer-to-peer interactions using technologies [24]; for Health and Wellbeing, mediating technology use by young children for optimal physical activity in the early years [25]; for Citizenship, the effective provision of online-safety education for young children [26]; and for Play and Pedagogy, the range of digital play activities provided to young children according to technology access in the classroom [27].

The sub-research question for each Investigation are as follows:(1)Relationships: What characterises infant and toddler peer-to-peer interactions using digital technologies?(2)Health and Wellbeing: How do families mediate technologies for optimal physical activity in the early years?(3)Citizenship: Does play-based learning about the internet help prepare children for later learning about online-safety?(4)Play and Pedagogy: How does classroom access to technologies influence educator provision of digital play activities?

### 2.3. Participants

The participants in this project include the Industry partners, the adults responsible for the education and care of young children (educators or parent/guardians), and young children. All four investigations use purposeful sampling [28] of the adults and children. The importance of involving children in the development of solutions to support them is well recognised [29]. In this project children will participate in the Exploration and Discovery phases of each investigation’s design and redesign cycles. For the Relationships Investigation, participants are drawn from Family Day Care settings; for Health and Wellbeing from Playgroups; for Citizenship from Long Day Care; and for Play and Pedagogy from Kindergartens. The industry partners participate in at least one investigation each according to their target audience and strategic objectives.

### 2.4. Ethics Approval and Consent to Participate

This project will abide by the World Medical Association Declaration of Helsinki [30] and has been approved by the Australian Catholic University Human Research Ethics Committee (Ref number: 2020-121H). All adult participants will provide written consent for their own participation and written consent for participation of children for whom they are guardians. Children aged three to five years provide assent using accepted conventions for researching with young children, including a child-centred explanatory statement using visual images to depict participation in interviews, video-recordings and/or the taking of photographs concerning their use of digital technologies [31]. For children aged birth to three years, the notion of ‘ethical symmetry’ in which researchers pay careful attention to children signaling disengagement or distraction (e.g., vocalizations, moving towards another activity) during periods of video-recording and/or photographic documentation will be used [32].

### 2.5. Sample Size

The central unit of analysis for this project is the practice generated by adults with children. Thus the sample sizes for each of the components of the project have been selected to ensure a robust capture of the practices children, families and educators enact and share according to the four areas of investigation.

### 2.6. Materials

Each investigation uses a specific measure. These are deployed at three time-points relative to the participatory design approach. The time-points are: T1 Pre-design; T2 Post-design; and T3 Post-Redesign. The measures were selected to capture evidence of the practices related to each Investigation (e.g., peer relationships; physical activity; internet understanding; technology access). The measures used for each investigation are: Relationships–A peer interaction observation protocol (adapted from [33]); Health and Wellbeing–Time-Use Diary and preschool activity patterns [34]; Citizenship–Children’s understanding of the internet interview [35]; Play and Pedagogy–Technology Environmental Scan (adapted for post-2005 technologies, [36]).

Across all four Investigations video-observations (one hour each) of children engaged in trial practices in the design and redesign cycles will be conducted. Educators and/or caregivers will also be invited to take photographs with children of trial digital practices. A semi-structured child-centred interview will be conducted with 15 children each in the Citizenship and Play and Pedagogy Investigations in the design and redesign cycles to understand their perspectives concerning valued digital practices. This will include the opportunity for children to reflect on what they like or dislike about documented trial practices (e.g., using videos and photographs as stimulus interview material). For the Relationships and Health and Wellbeing investigations, opportunities for infants and toddlers to engage with a variety of technologies will be provided, and children’s interest and/or lack of indicated interest in these documented. This will include observations of children’s vocalizations and movements when engaging with the technologies. This will involve up to 15 infants and toddlers each in the Relationships and Health and Wellbeing investigations.

Table 1 provides an overview of the four investigations relative to the design and redesign cycles, including sub-research question, participants, industry partners, measure and brief description of procedure.

## 3. Process and Analysis

### 3.1. Design and Redesign Cycles

Figure 1 provides an overview of the phases within the design and redesign cycles conducted by each of the four investigations. The design cycle focusses on enacting practices and is conducted relatively independently by each investigation. During the redesign cycle there are explicit processes for sharing practices across the four investigations.

#### 3.1.1. Years 1–2 Design Cycle (Exploring, Discovering, Prototyping)

Exploring: This phase begins with industry partner staff-members in each of the four investigations conducting an audit of resources, materials and/or programs already available within their organisation relevant to the investigation in which they are participating. These organisational items are then shared with the researchers leading each investigation by the industry Partner staff-members. Together the researchers and industry partner staff-members match the items to the available practice-advice in the ECA Statement. This matching activity precedes Workshop 1. During Workshop 1 industry partner staff members, the researchers and adult participants from each Investigation come together. Each Workshop follows the same protocol. First, the researcher conducts a guided-reading of the ECA statement practice-advice with participants relevant to their area. Second, the researcher leads a discussion with participants and the industry partner staff members regarding the substantive issue comprising their investigation. Third, the industry partner staff-members share with participants items from their organisation matched with the ECA statement practice-advice relevant to their investigation. Fourth, the researcher leads a brainstorming session with participants and the industry partner staff-members responding to the issue [using the matched ECA statement practice-advice and industry partner items]. Fifth, the researcher supports the participants to identify actions and interactions from the brainstorming session of potential value for addressing the issue within their respective setting. Sixth, the participants and industry partner staff-members agree [with guidance from the researcher] on one or more valued actions/interactions for trial implementation in the Discovering Phase.

Discovering: Identified actions/interactions from the earlier Exploring phase are **trialled** by the participants according to the nominated procedure for their investigation (Table 1). These are implemented over a 12-week period [37] with trial experiences **documented** by industry partners, participants and researchers. Industry partner staff-members from each investigation maintain regular contact with participants by phone, email, and/or video-conferencing working alongside researchers. This contact facilitates participant access to industry partner items relevant to the investigation area. Industry partner staff-members keep an audio-diary during this period, noting any participant request for item access, challenges and/or benefits associated with implementation of their trial actions/interactions. Researchers from each Investigation send one text message per week to participants, inviting photographic documentation of the trial actions/interactions. Photographs are returned to a dedicated email address for each Investigation, retrieved and stored on a dedicated site. Researchers conduct one-hour video-observations of the trial actions/interactions enacted by participants in their investigations. After 12 weeks, a semi-structured interview is conducted with adult participants in each investigation. Where applicable researchers also complete a child-centred interview using techniques appropriate for researching with children of various ages [38].

Prototyping: Researchers collate the data documented during the Discovering Phase, including the audio diaries, photographs, video-observations, adult and child-centred interviews from all four investigations. Researchers deductively analyse photographs and video-observations for ‘actions and interactions’ [9], and diaries and interviews for ‘doings and sayings’ [11]. Coded data are paired, linking actions and interactions with doings and sayings enabling the identification of enacted practices. All coding and pairing are checked by another researcher for inter-rater reliability. Paired data are then extracted from the Discovering phase data set as exemplar practices. Two researchers work with the Australian Broadcasting Corporation and Deeper Richer to translate the exemplars into content objects as digital exemplars. Digital exemplars represent the practices as videos, visualizations, animations, memes, cartoons, and/or infographics. These formats are deliberately used because they do not require identifying images of adult and/or child participants (e.g., from video-observations or photographs) to be used at any stage.

#### 3.1.2. Year 3 Redesign Cycle (Exploring, Discovering, Prototyping)

Exploring: The second Exploring phase commences with industry partner staff-members, researchers and adult participants from all four investigations attending a shared webinar. The webinar is hosted and recorded by Raising Children Network. During the webinar, each investigation researcher showcases the digital exemplars generated from their investigation as per the Year 1–2 Prototyping Phase. A live-chat led by a member of the research team during the webinar provides opportunity for the industry partner staff-members and participants to identify digital exemplars representing valued practices for application in their own Investigations. After the Webinar, all digital exemplars are uploaded to a Learning Management System (LMS) supported by ECA. For a period of two-weeks post-webinar, industry partner staff-members, researchers and participants continue to discuss and share the digital exemplars across all four investigations via the LMS. The Webinar and LMS discussion precede Workshop 2. During Workshop 2, Industry partner staff members, researchers and participants return to their respective investigation, repeating the workshop protocol used in Workshop 1–this time drawing on the shared digital exemplars from the Webinar and LMS discussion to identify valued practices of use for their investigation.

Discovering and Prototyping: The Discovering phase and Prototyping phase of the redesign cycle are then conducted in the same manner as these phases in the initial design cycle, with shared actions/interactions trialled and documented and content objects created. Finally, at the end of year 3 the industry partner staff members, researchers and participants from all four investigations participate in a second webinar and LMS discussion.

### 3.2. Year 4 Building the Online Tool

T1 and T2 and T3 data generated in each Investigation are analysed using a combination of descriptive statistics and content analysis. For T1 and T2 any established differences are used to identify *enacted* practices and for T2 and T3 to identify *shared* practices [10]. All shared practices are mapped back to their point of origin as enacted practices (e.g., from the Discovering phase of the first design cycle). This mapping will be converted into a visual representation of all enacted and shared practices over the course of the design and redesign cycles. This visual representation creates the field of action intelligibility answering the primary research question: What practices do children, families and educators enact and share according to the four areas of practice-advice provided in the ECA Statement on Young Children and Digital Technologies?

Nodes indicated in the visual field of intelligibility represent the practices at their point of origin, while connections drawn between the nodes illustrate shared practices (Figure 2). Valued practices within the field are those enacted and shared amongst the children, families and educators from the diverse situations represented within each Investigation (Relationships–Family Day Care; Health and Wellbeing–Playgroups; Citizenship–Long Day Care; Play and Pedagogy–Kindergarten).

Deeper Richer works with two of the researchers to convert the field of action intelligibility as visually represented into the online tool in the form of an interoperable database hosting the digital exemplars at each node. The online tool is developed during two rounds of ‘build, test, refine’. Data analytics using normalized frequency counts are applied to illustrate the strength of connection between nodes as the industry partners access the online tool during these rounds [37]. Strength of connection is visibly represented in the online tool as various degrees of line thickness between nodes (e.g., thicker lines indicating a stronger degree of connection). An in-built function within the online tool allows new digital exemplars to be added to the field of action intelligibility and/or for new connections between nodes to be indicated.

Participants and industry partner staff-members from each investigation participate in the ‘Testing’ aspect developing the online tool. By the end of round two, the online tool is operational as a minimally viable product. The researches will lead an implementation seminar working with the industry partners using the online tool to realise their strategic objectives.

## 4. Discussion

It is internationally recognised that young children are growing up in digital society [5]. This recognition is reflected in the number of guidelines currently advising families and educators on technology use by young children. While intended to help adults make decisions in the best interests of children, many of these guidelines promote messaging that is inconsistent [39]. For example, the Australian 24-Hour Movement Guidelines for the Early Years [40], the American Academy of Pediatrics [2], and the World Health Organisation [3] all suggest screen-based technologies should not be used with children aged birth to two-years, and only for one hour per day with children aged 3–5 years. In contrast, the UK Royal College of Pediatrics and Child Health [4] recommend that parents and professionals regulate technology use by children relative to family circumstances, including the influence of technologies on children’s snacking habits, sleep and participation in non-digital activities. Alternatively, the European Academy of Paediatricians and European Early Childhood Obesity Group [41], recommend both restriction and relativity–suggesting technologies be used for no more than 1.5 h per day by children aged 4 years and over; that educators promote human interaction over engagement with digital technologies; and finally, that families co-use technologies with their children. Such mixed-messaging from prominent organisations is not only confusing for families and educators but provides an inadequate knowledge base for the Industry Partners realising the best interests of young children. It is not entirely clear which stance the Industry Partners themselves should take–restriction, relativity or a combination of both?

This project provides a way forward by focusing on practices in digital society. Practices help shift attention away from technological determinist attempts at managing the impact of technologies on children from a socially causative perspective, towards understanding what people do in digital society, and why. This shift from determinism to practices is significant because understanding what people do in digital society (and why) helps orientate research, practice and policy towards helping young children and their families live with technologies for the best possible health and educational outcomes, rather than seeking to advise on levels of technological restriction that are not always socially feasible, e.g., moving past recommended ‘screen time’ limits for young children during COVID-19.

Critical constructivism makes an important contribution to understanding practices in digital society because it highlights how human values are always evident in technological innovation and use [14]. Digital society is not a static situation in which adults must manage technologies on behalf of young children, but a dynamically responsive situation in which young children and their adults use technologies for multiple purposes. The identification of valued practices for young children growing up in digital society, as enacted and shared amongst children, families and educators across different situations forecasts human agency in the deliberate use of technologies in the best interests of young children. This project therefore engages the industry partners and researchers in a planned response to the problem of mixed-messaging in adult-decision making about technology use ‘with, by and for’ young children.

Working to integrate four Investigations using participatory design, this project operates according to an established timeline of activity, in which all T1, T2 and T3 measures are conducted simultaneously by each Investigation. This is also the case for all Industry Partner and participant involvement in Workshops, Webinars and periods of trialling practices. Operationally, the complexity of this joint timing is managed through regular meetings between lead Investigation researchers, reporting on participant recruitment, scheduling of measures, data generation and workshop design and implementation. Researchers also report bi-annually to an international advisory board established for this project, focussing on the intersection between Investigations and the broader remit of the project regarding the identification of the practices enacted and shared amongst children, families and educators from across all four Investigations. Practicalities associated with multiple forms of data generation are managed via a central project depository for all measures, video-observations, photographs and/or interviews.

## 5. Conclusions

Young children, aged birth to five years are growing up in digital society. Digital society comprises the invention and use of technologies by people for generating, storing, sharing and communicating information in digital form. According to critical constructivism, human values manifest in the design and use of technologies by people over time. Practice theory shows how people enact and share actions and interactions of value to them in various social situations. This project is predicated on the argument that identifying the practices enacted and shared amongst young children and their adults in different social situations (e.g., Family Day Care, Playgroups, Long Day Care, Kindergarten) will provide insight into how and why digital technologies may be used ‘with, by and for’ young children for optimal health and educational outcomes. The industry partners in this project will benefit from access to the online tool, featuring exemplar digital practices as an informant to their own service provision reaching children, families and educators. It is intended that the online tool will help mediate the impact of mixed-messaging from existing guidelines regarding young children and digital technologies in the deliberate use of technologies in the best interests of young children’s health and educational outcomes.

## Figures and Tables

**Figure 1 ijerph-17-08819-f001:**
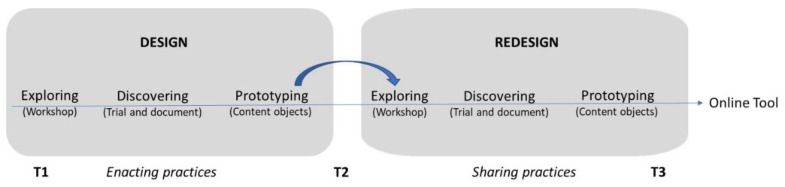
Phases of the design and redesign cycles.

**Figure 2 ijerph-17-08819-f002:**
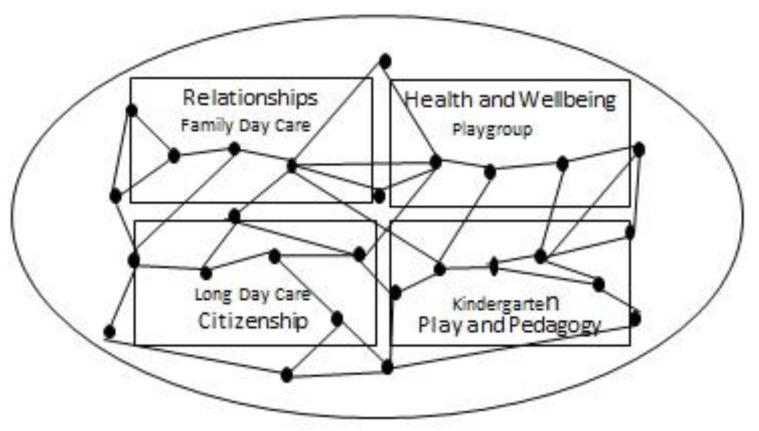
Field of action intelligibility.

**Table 1 ijerph-17-08819-t001:** Four investigations relative to design and Redesign cycles.

Investigation	Sub Research Question	Participants	Industry Partners	Measure	Procedure
Relationships	What characterises infant and toddler peer-to-peer interactions using digital technologies?	8 Family Day Care Educators [Certificate III] and 40 children aged 8–36-months [5 per Family Day Care Educator].	Early Childhood Australia; Raising Children Network	Peer Interaction Observation Protocol (adapted from Engdahl [33])	Family Day Care Educators participate in Design and Redesign cycles, implementing trial actions and interactions during the Discovering phase of each cycle. CI attends one implementation of trial actions per Family Day Care Educator to capture ‘rich description’ [via fieldnotes] [33] of peer-to-peer interactions.
Health and Wellbeing	How do families mediate technologies for optimal physical activity in the early years?	16 Playgroup representatives and 40 families with children aged 18-months [5 families per Playgroup].	Australian Broadcasting Corporation; Raising Children Network	Time-Use Diary [34]	Playgroup families participate in Design and Redesign cycles. Playgroup representatives participate in national Webinars. Children followed from 18-to-24-months; and 24-months-to-36-months.
Citizenship	Does play-based learning about the internet help prepare children for later learning about online-safety?	8 Long Day Care Educators [Diploma to Bachelor Education] and 80 children aged 2–5 years [10 children per Educator].	Australian Federal Police; Alannah and Madeleine Foundation; eSafety Commissioner	Children’s understanding of the internet interview schedule [35]	Teachers and children assigned to Intervention or Control group (5 Teachers; 100 children per group). Intervention Teachers participate in Design and Redesign cycles implementing trial actions during the Discovering phase of each cycle. 50 Intervention and 50 Waitlist children followed into school offering AFP ThinkUKnow and/or AMF eSafety Schools during Year 3.
Play and Pedagogy	How does classroom access to technologies influence educator provision of digital play activities?	8 Kindergarten Teachers [Bachelor of Education] and 160 children aged 3–5-years [twenty children per Teacher].	Early Childhood Australia; Deeper Richer	Technology Environmental Scan [adapted to include post-2005 technologies] [36]	Educators work in high/low classroom technology access pairs during the Design and Redesign cycles. Educators implement trial actions and interactions during the Discovering phase of each cycle shadowing each other in the classroom for at least two hours each.

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
