# Peer review of "Rationale, Design and Methods Protocol for Participatory Design of an Online Tool to Support Industry Service Provision Regarding Digital Technology Use ‘with, by and for’ Young Children"

_ijerph, 2020, doi:10.3390/ijerph17238819_

Round 1
Reviewer 1 Report
This article addresses a very innovative theme associated with the messages received by adults who educate and care for young children are exposed to contradictory messages about what is best for young children in the digital society. These mixed messages make it difficult for adults to make decisions about using technology in the best interests of young children.
The topic addressed by the article is very interesting and addresses a very appropriate methodology. This article addresses this issue by working with leading organizations that provide services related to quality digital media production, online safety education, digital gaming, and digital parenting.
The methodology used has been Participatory Design. It has involved researchers, industry partners, and end users (eg, children, families, and educators) collaborating on a problem with the intention of creating a solution.
It would be necessary for the authors to explain the phases followed in a clearer way for the reader, for example, through a figure. The figure could show the sequence of work followed and the actors involved.
Participatory Design operates in repeated Design and Redesign cycles. Each cycle consists of three phases. These are: 1) Exploring; 2) Discovering; and 3) Prototyping.
In the same way, it would be necessary to explain what happens in each phase and argue in a more consistent theoretical way.
On the proposed objectives: to identify the practices enacted and shared among children, families and educators in the digital society to create an online tool for the provision of services with empirical information on the use of digital technology "with, by and for "young children.
It is recommended that the objective of the study should be more concrete and go in line with the research questions.
They address the investigated topic correctly and subsequently guide the presentation of the conclusions.
Results are clearly presented and properly analyzed. As it is such a complex study, it is necessary to explain each phase in a concrete way so that the reader can see where the study is at and the results obtained from that phase of the investigation. The article could be improved with graphics associated with methodology and results.
Conclusions and discussion are supported by existing theory on the subject and previous research.
The research and practical implications are clearly identified in the article.
The format of the research questions must be reviewed as it is not correct. The same happens with bibliographic references that should be reviewed.
Author Response
Author Responses to Comments and Suggestions for Authors (reviewer comments quoted first, with author responses in bullet points)
This article addresses a very innovative theme associated with the messages received by adults who educate and care for young children are exposed to contradictory messages about what is best for young children in the digital society. These mixed messages make it difficult for adults to make decisions about using technology in the best interests of young children.
- Thank you for acknowledging the innovation and the important societal problem it addresses.
The topic addressed by the article is very interesting and addresses a very appropriate methodology. This article addresses this issue by working with leading organizations that provide services related to quality digital media production, online safety education, digital gaming, and digital parenting.
The methodology used has been Participatory Design. It has involved researchers, industry partners, and end users (eg, children, families, and educators) collaborating on a problem with the intention of creating a solution.
- Thank you for your kind comments on the methods being appropriate.
It would be necessary for the authors to explain the phases followed in a clearer way for the reader, for example, through a figure. The figure could show the sequence of work followed and the actors involved.
- Thank you for this suggestion to clarify the phases with a figure. We have created a new Figure 1.0 to clarify this.
Participatory Design operates in repeated Design and Redesign cycles. Each cycle consists of three phases. These are: 1) Exploring; 2) Discovering; and 3) Prototyping.
In the same way, it would be necessary to explain what happens in each phase and argue in a more consistent theoretical way.
- In addition to the new figure created in response to the previous review comment, we have also edited the text to provide greater clarity around the different phases for each cycle (see lines 242 and 312)
On the proposed objectives: to identify the practices enacted and shared among children, families and educators in the digital society to create an online tool for the provision of services with empirical information on the use of digital technology "with, by and for "young children.
It is recommended that the objective of the study should be more concrete and go in line with the research questions.
They address the investigated topic correctly and subsequently guide the presentation of the conclusions.
- Thank you for this suggestion, we have substantially edited the text to clarify the aim and objectives and align them more closely with the research questions (see lines 135 and 144).
Results are clearly presented and properly analyzed. As it is such a complex study, it is necessary to explain each phase in a concrete way so that the reader can see where the study is at and the results obtained from that phase of the investigation. The article could be improved with graphics associated with methodology and results.
- Thank you for the positive comments, and our changes to clarify the phases are detailed above.
Conclusions and discussion are supported by existing theory on the subject and previous research.
- Thank you for your positive comments.
The research and practical implications are clearly identified in the article.
- Thank you for your positive comments.
The format of the research questions must be reviewed as it is not correct.
- Thank you for picking up this error. We have corrected the formatting.
The same happens with bibliographic references that should be reviewed.
- We have corrected the reference formatting also.
Reviewer 2 Report
I enjoyed reading such a well-written article detailing a thoroughly articulated protocol for an interesting project with clearly defined goals and a much needed and very useful output. While the development of the online tool has a very clear practical application, the research is also well grounded theoretically with a novel approach. I particularly liked the researchers taking an approach that challenges a deterministic view of digital technologies. This is an exciting project strengthened by the collaboration between academics, researchers and industry stakeholders. The research design and methods are described in impressive detail providing a coherent overview of the entire 4-year project.
However, given this project is about practices 'with, by and for young children' I would have liked more detail about children’s participation in the project and ethical considerations in relation to this. The process and analysis section mostly relates to adult participants. A child-centred interview is mentioned, but it was not clear what this would cover (or was this the Children's understanding of the internet interview schedule in the Citizenship Investigation?). Also, as stated at line 255, this interview would be completed with up to 15 children (aged 3-5 years) per Investigation. However, it would seem that this would not be possible for each Investigation as the ages of the children in the Health and Wellbeing and Relationship Investigations did not include this age group?
Furthermore, while Pascal and Bertram's article about children's active participation and voice in research is cited as guiding your research techniques, how children's active participation and perspectives might be obtained and contribute to the development of the online tool is not clear. Therefore, the paper could be improved by providing more information about if, and how, the project’s participatory design involves children as active participants (as opposed to being the subjects of adult observation) and the ethics around consent/assent processes that will be employed in each setting.
Author Response
Author Responses to Comments and Suggestions for Authors (reviewer comments quoted first, with author responses in bullet points)
I enjoyed reading such a well-written article detailing a thoroughly articulated protocol for an interesting project with clearly defined goals and a much needed and very useful output. While the development of the online tool has a very clear practical application, the research is also well grounded theoretically with a novel approach. I particularly liked the researchers taking an approach that challenges a deterministic view of digital technologies. This is an exciting project strengthened by the collaboration between academics, researchers and industry stakeholders. The research design and methods are described in impressive detail providing a coherent overview of the entire 4-year project.
- Thank you for your positive comments on the article being well written and thoroughly articulated. We also appreciate your acknowledgement of the very practical work being well grounded theoretically.
However, given this project is about practices 'with, by and for young children' I would have liked more detail about children’s participation in the project and ethical considerations in relation to this. The process and analysis section mostly relates to adult participants. A child-centred interview is mentioned, but it was not clear what this would cover (or was this the Children's understanding of the internet interview schedule in the Citizenship Investigation?). Also, as stated at line 255, this interview would be completed with up to 15 children (aged 3-5 years) per Investigation. However, it would seem that this would not be possible for each Investigation as the ages of the children in the Health and Wellbeing and Relationship Investigations did not include this age group?
Furthermore, while Pascal and Bertram's article about children's active participation and voice in research is cited as guiding your research techniques, how children's active participation and perspectives might be obtained and contribute to the development of the online tool is not clear. Therefore, the paper could be improved by providing more information about if, and how, the project’s participatory design involves children as active participants (as opposed to being the subjects of adult observation) and the ethics around consent/assent processes that will be employed in each setting.
- Thank you for the suggestion to provide more detail about the nature of the participation of children in this project. We are pleased to provide more detail (see lines 188, 201, 223)
- Regarding the ethical considerations of the children’s involvement we have also included more detail on this (see lines 201)